# Fire Propagation Characteristics and Fire Risks of Polyurethanes: Effects of Material Type (Foam & Board) and Added Flame Retardant

Ji Hun Choi [1] , Seung Un Chae [2], Euy Hong Hwang [1] and Don Mook Choi [1,*]

1   Department of Equipment System, Fire Protection Engineering, Gachon University, 1342, Seongnam-daero, Sujeong-gu, Seongnam-si 13120, Korea; coolpishun@gachon.ac.kr (J.H.C.); dmlghd2@gachon.ac.kr (E.H.H.)
2   Department of Fire Safety Research, Korea Institute of Civil Engineering and Building Technology, 64, Mado-ro 182beon-gil, Mado-myeon, Hwaseong-si 18544, Korea; seungun.chae@kict.re.kr
*   Correspondence: fire@gachon.ac.kr; Tel.: +82-31-750-5716; Fax: +82-31-750-8749

**Abstract:** Polyurethane material is used as an interior finish and wall cavity insulation. Flame-retardant products may be used for ignition, flame diffusion, and heat-release blocking. A large-scale test was conducted to understand the flame propagation characteristics of polyurethane with the addition of a flame retardant. The fire propagation properties and fire risks of four commonly used polyurethane materials were examined using three tests. Specifically, ignition properties, flame propagation behavior, and flashover occurrence were probed using full-scale tests, while heat release and fire characteristics were examined using cone calorimeter tests, and the toxicity of gaseous combustion products was assessed using gas toxicity tests. PIR F and PIR B, which contained flame retardants, featured slow flame propagation and a long-lasting residual flame, and PIR F released HCl and $Br_2$ on combustion. Full-scale tests revealed that although external flame propagation was always accompanied by flashover, irrespective of whether the flame retardant was present, a delay or blockage of energy transfer to the inside was observed for flame-retardant-containing specimens. Apart from checking the safety at the material level, the importance of identifying the actual fire characteristics through a full-scale test was confirmed.

**Keywords:** polyurethane resin; polyisocyanurate resin; flame propagation characteristics; fire risk; real fire test; foam; board; flame retardant; gas toxicity test; cone calorimeter test





## 1. Introduction

Polyurethanes are widely used in interior/exterior finishings, construction panels, composite material cores, and frozen warehouse and factory construction, owing to their low density, high compression strength, low thermal conductivity, superior cohesive force, and low absorption rate [1]. They are manufactured in the form of either foams or boards (Figure 1) [2]. Polyurethane has a very low water absorption rate. When an insulation material absorbs water, the thermal conductivity of the material is affected, and this, in turn, adversely affects the insulation effect. Rigid polyurethane foam exhibits a low density and superior thermal insulation performance, thus suiting applications in roofing, frozen warehouse construction, flame-retardant composites, water storage, fillers, and panels. Meanwhile, polyurethane boards are used as thermal insulation materials for the walls, floors, and roofs of buildings, sandwich panel cores, and automobile interiors [3].

However, polyurethanes are very sensitive to heat and suffer from rapid ignition and fire propagation, as well as foam collapse [4], which make early-stage polyurethane fire quenching practically impossible. In Korea, hot works (e.g., welding) performed in the immediate vicinity of heat insulation materials with poor thermal stability (e.g., urethane products) and without any protective measures have been triggering an

increased number of unexpected large-scale fires during the construction of frozen ware-houses and other buildings [5]. According to the Korean Ministry of Public Administration and Security, 5829 fires broke out during welding, cutting, or grinding works during 2015–2019, resulting in 444 casualties (32 dead and 412 injured). The flammable gases produced when polyurethane burns have very high flame-propagation speeds. Yet, it is possible to delay flame propagation by coating the surface with a non-combustible or flame-proof inorganic material. As such, safe handling and utilization of polyurethanes require a deep understanding of their ignition properties and other fire risk-related characteristics.

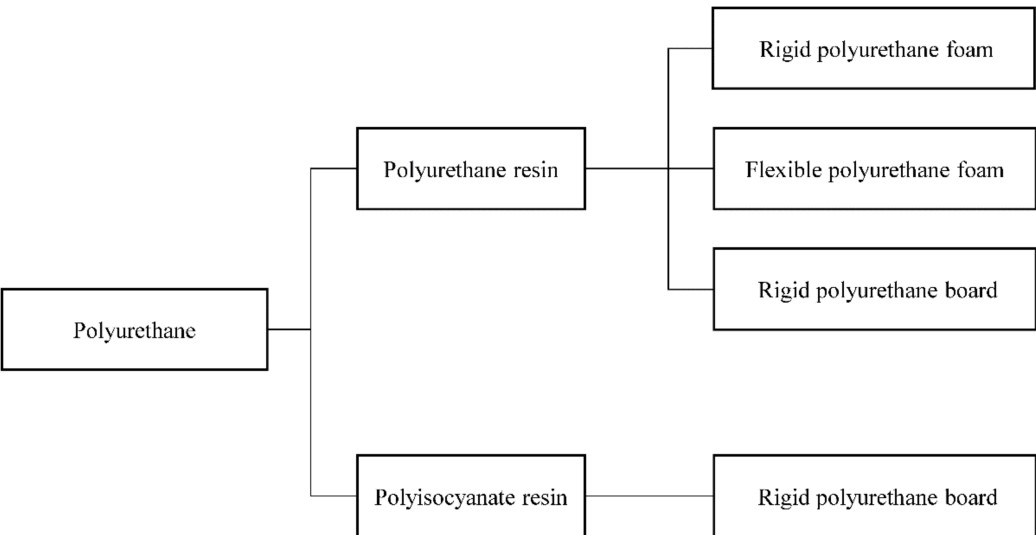

**Figure 1.** Classification of polyurethanes.

When polyurethane foam is used as insulation, either in cavities or exposed as an interior finish, it is subject to heavy regulation throughout the world. The reason for this is that there is a significant amount of experience showing that foam plastics can yield misleading results in small- or intermediate-scale tests. For example, many foam plastics perform well in fire tests where they can recede or melt and thus avoid exposure to the actual ignition source during the test [6].

In other studies, fire tests using small-scale specimens were performed to determine the ignition and heat release rate (HRR) of polyurethane; however, the flame propagation characteristics of the full-scale fire test were not considered.

In this work, the ignition properties, fire propagation behaviors, and flashover occurrence characteristics of common polyurethane materials were quantitatively analyzed using full-scale tests, while the related dissipation and fire characteristics were examined using cone calorimeter tests, and the toxicities of combustion products were assessed using gas toxicity tests.

## 2. Materials and Methods

Herein, among the polyurethane insulation materials used in South Korea, we selected four foams and boards, namely polyurethane resin foam (PUR F), polyisocyanurate resin foam (PIR F), polyurethane resin board (PUR B), and polyisocyanurate resin board (PIR B), and evaluated their fire risks using three types of tests.

Cone calorimeter tests were used to determine the maximum, average, and total heat-release rates as metrics of thermal characteristics, while gas toxicity tests were used to measure the average incapacitation times of mice and thus evaluate the toxicity of gaseous combustion products generated in the presence of a flame retardant. Finally, full-scale mockup tests were used to examine the flame propagation speed, specimen temperatures at different locations, and flashover occurrence.

### 2.1. Materials

As shown in Table 1, for the polyurethane used in this test, a product conforming to KS M 3809 was selected [7].

**Table 1.** Polyurethane foam and board physical properties.

| Material | Density | Thermal Conductivity | Flexural Fracture Load | Compressive Strength | Moisture Absorption |
|---|---|---|---|---|---|
| | (kg/m$^3$) | (W/m·K) | (N) | (N/cm$^2$) | (g/100 cm$^2$) |
| PUR F | | 0.024 or less | | 20 or more | |
| PIR F | 35 or more | | 25 or more | | 3 |
| PUR B | | 0.023 or less | | 10 or more | |
| PIR B | | | | | |

Polyethylene processed paper specified in KS T 1037 is used for the face material of the PIR B shell, and the adhesive aluminum specified in KS D 9003 is otherwise used.

PIR is synthesized by the trimerization of isocyanate molecules to form isocyanurate rings, which feature excellent thermal stability and fire resistance and also become the points at which the polymer matrix achieves cross-linking. As isocyanurate rings decompose at higher temperatures (450–650 °C) than urethane bonds (200–350 °C), PIR can better withstand high temperatures than PUR [8]. Accordingly, in the case of PIR, a two-stage decomposition featuring the disintegration of urethane bonds and isocyanurate rings is observed. The activation energies of urethane and isocyanurate generation equal 38.7 and 60.3 kcal mol$^{-1}$, respectively [9].

Although PUR has a linear structure that is structurally vulnerable and poorly resistant to heat because of the absence of flame retardancy, it features superior thermal conductivity compared to other thermally insulating materials. PIR is widely used in the form of boards but is difficult to produce in the form of foam as it is prone to sudden-onset reactions. Moreover, the stable annular structure of the isocyanurate ring endows PIR with high flame retardancy and resistance to temperature change. Thus, PIR is superior to PUR in terms of fire risk as the minimum ignition energy required for the temperature change and combustion reaction is larger in the former case. Although the endothermic carbonization of the PIR surface during combustion is feasible, the generated char decelerates the progress of ignition toward the inside. However, a fire risk still exists when rapid combustion on the surface promotes ignition.

As shown in Table 2, the material composition and content of the polyurethane used in the test are as follows. In the case of polyurethane foam and board, there is a difference in the manufacturing method, but the composition of the material is the same and there is no significant difference in the mixing ratio.

### 2.2. South Korean Standards Related to Insulating Materials

According to Article 61 of the Enforcement Decree of the Building Act and Article 24 of the Building Fire Protection Construction Rules [10], the external finishings of buildings with three or more floors should contain flame retardants or higher-grade materials (including insulating materials). Although flame retardants or higher-grade interior finishing materials should be used for the construction of warehouses with a total floor area of ≥600 m$^2$ or factories with an area of ≥1000 m$^2$, as internal insulation is currently excluded from the management objects, insulating materials not satisfying flame-retardant ratings are still thoughtlessly used. Board-type urethane foam products, as construction finishing materials, can be included in insulating materials, while spray-type urethane foam products are classified as spray-coating materials and not as insulating materials.

**Table 2.** Polyurethane foam and board composition and content.

| Composition | CAS Number | Content (%) | |
|---|---|---|---|
| | | **PUR F and B** | **PIR F and B** |
| Isocyanic acid polymethylenepolyphenylene ester; polymethylene polyphenylene isocyanate | 9016-87-9 | - | - |
| Diphenylmethane 4,4′-diisocyanate | 101-68-8 | - | - |
| Diethylene glycol | 111-46-6 | 5~15 | 5~15 |
| 1,1′-Dichloro-1-fluoroethane (HCFC 141b) | 1717-00-6 | 20~30 | 20~30 |
| Tris(1-chloro-2-propyl) phosphate | 13674-84-5 | 5~15 | 10~20 |
| α-Hydro-ω-hydroxypoly[oxy(methyl-1,2-ethanediyl)] | 25322-69-4 | 55~65 | 50~60 |
| Etc(trade secret) | - | 1~10 | 1~10 |

Non-combustibility, semi-non-combustibility, and flame retardancy are evaluated using KS F 1182 (non-combustibility test), KS F ISO 5660-1 (cone calorimeter test), and KS F 2271 (gas toxicity test) standards. In the case of composite finishing materials such as sandwich panels, the risk of fire is complexly evaluated through the analysis of the heat release rate, flame propagation rate, and smoke production rate using the KS F ISO 13784-1 standard (medium-scale combustion test of sandwich panels). Table 3 lists the grades of flame retardancy and semi-non-combustibility as well as their evaluation methods and performance standards.

**Table 3.** Performance criteria of non-combustible, semi-combustible, and flame-retardant materials.

| Material Class | Standard | Evaluation Criteria |
|---|---|---|
| Non-combustible | KS F ISO 1182 (combustibility testing ofarchitectural materials) | Maximum temperature after 20-min heating should not exceed the final equilibrium temperature of 20 K. When no equilibrium is reached in 20 min, the average time in the last 1 min is considered as the final equilibrium time. The nitrogen reduction rate should be equal to or less than 30%. |
| | KS F 2271 (gas toxicity test) | Average incapacitation time of laboratory mice ≥9 min. |
| Semi-non-Combustible | KS F ISO 5660-1 (cone calorimeter test) | Total radiant heat after 10-min heating should not exceed 8 MJ m$^{-2}$. Within 10 min, the maximum heat radiance rate should not exceed 200 kW m$^{-2}$ for longer than 10 consecutive seconds. No sample-penetrating cracks, holes, or melting (for mixed-content materials, this term includes the melting and dissipation of all core materials) should be observed after 10-min heating. |
| | KS F 2271 (gas toxicity test) | Average incapacitation time of laboratory mice ≥9 min. |
| Fire-retardant | KS F ISO 5660-1 (cone calorimeter method) | Total radiant heat after 5-min heating should not exceed 8 MJ m$^{-2}$. Within 5 min, the maximum heat radiance rate should not exceed 200 kW m$^{-2}$ for longer than 10 consecutive seconds. No sample-penetrating cracks, holes, or melting (for mixed-content materials, this term includes the melting and dissipation of all core materials) should be observed after 5-min heating. |
| | KS F 2271 (gas toxicity test) | Average incapacitation time of laboratory mice ≥9 min. |

*2.3. Evaluation of Combustion Properties*

The cone calorimeter test simulates actual fire conditions using specimens with dimensions of 100 × 100 × 50 mm and allows one to evaluate the ignition properties, heat release rate (HRR), fire propagation characteristics, and combustion gas toxicity of small single-material samples. The peculiarity of this test is that HRR is calculated using the oxygen consumption rate method, in which radiant heat is applied at a flux of 50 kW m$^{-2}$

using a cone-shaped heater, and the combustible gas generated from the specimen is ignited using an electric ignition source [11]. Figure 2 illustrates the test setup and the employed equipment.

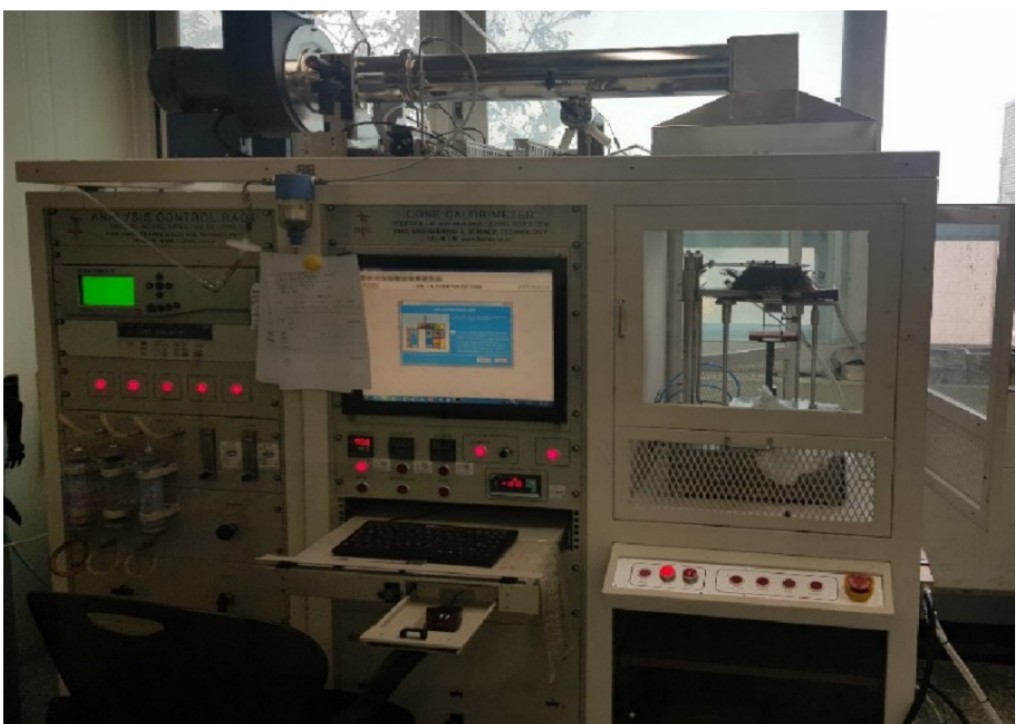

**Figure 2.** Setup of the cone calorimeter test.

The above test was conducted in triplicate, and the HRR was calculated as the corresponding average. Figure 3 shows the employed specimens, and Table 4 lists their properties.

### 2.4. Evaluation of Combustion Gas Toxicity

The test used to evaluate the toxicity of gaseous combustion products was performed in duplicate using 220 × 220 × 150 mm specimens with three 25-mm-diameter perforating holes. The thickness of the specimen should be the same as that of the actual product, so when the thickness exceeded 150 mm, it was reduced to 150 mm. The specimens were heated for 3 min using the ancillary heat source and then further heated for 3 min using the main heat source. Air was supplied only during heating using the primary and secondary supply devices of the heating furnace at rates of 3 and 25 L min$^{-1}$, respectively. Figure 4 illustrates the test setup and the employed equipment [12], while Figure 5 shows the employed specimens and Table 5 lists the properties of these specimens.

**Table 4.** Properties of the samples used for cone calorimeter tests.

|  | PUR F | | PIR F | | PUR B | | PIR B | |
|---|---|---|---|---|---|---|---|---|
|  | Mass (g) | Density (kg m$^{-3}$) | Mass (g) | Density (kg m$^{-3}$) | Mass (g) | Density (kg m$^{-3}$) | Mass (g) | Density (kg m$^{-3}$) |
| Test 1 | 19.70 | 39.40 | 19.40 | 38.80 | 23.10 | 46.20 | 27.30 | 54.60 |
| Test 2 | 19.60 | 39.20 | 18.90 | 37.80 | 21.10 | 42.20 | 28.60 | 57.20 |
| Test 3 | 19.70 | 39.40 | 20.40 | 40.80 | 23.20 | 46.40 | 29.20 | 58.40 |
| Average | 19.67 | 39.33 | 19.57 | 39.13 | 22.47 | 44.93 | 30.27 | 60.53 |

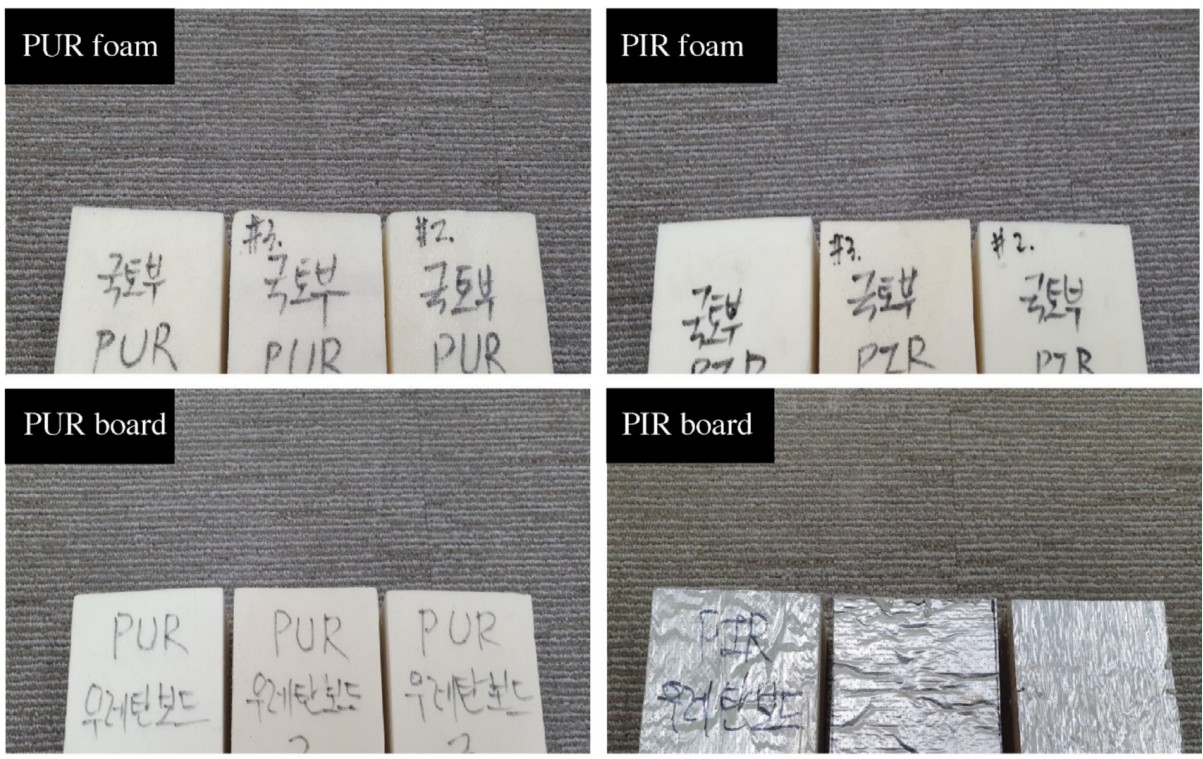

**Figure 3.** Samples used for cone calorimeter tests (four types).

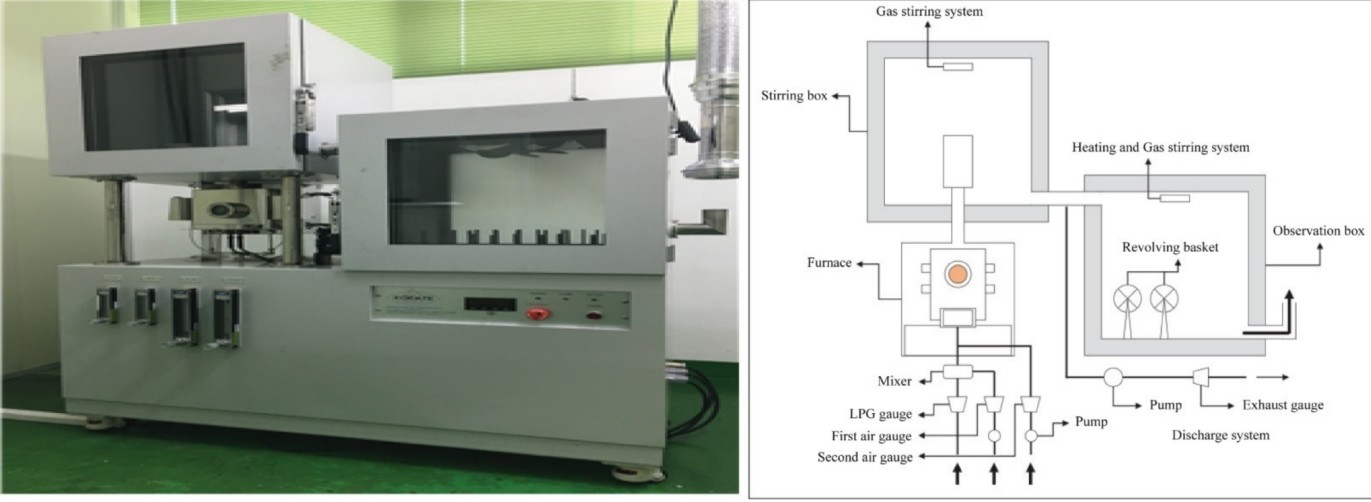

**Figure 4.** Setup of the gas toxicity test.

**Table 5.** Properties of the specimens subjected to gas toxicity tests.

| | PUR F | | PIR F | | PUR B | | PIR B | |
|---|---|---|---|---|---|---|---|---|
| | Mass (g) | Density (kg m$^{-3}$) | Mass (g) | Density (kg m$^{-3}$) | Mass (g) | Density (kg m$^{-3}$) | Mass (g) | Density (kg m$^{-3}$) |
| Test 1 | 144.90 | 30.00 | 150.70 | 31.07 | 175.60 | 35.99 | 206.90 | 42.58 |
| Test 2 | 139.10 | 28.80 | 162.40 | 33.45 | 173.00 | 35.46 | 205.80 | 42.44 |
| Average | 142.00 | 29.40 | 156.55 | 32.26 | 174.30 | 35.73 | 206.35 | 42.51 |

The emission of gases by the discharge device of the tested box was conducted only during heating, and the emission rate was set to 10 L min$^{-1}$. The temperature of the exhaust gas was measured using a specified thermocouple and a thermometer. The temperature in

the tested box initially equaled 30 °C, and the combustion gas was flown into the tested box containing five-week-old DD or ICR female mice (18–22 g) in rotating baskets. The average incapacitation times of eight mice were calculated as

$$X = {}^{\overline{}}X - \sigma \tag{1}$$

where X is the average activity stop time (min), ⁻X is the average value of the behavioral stop time of 8 mice (min), and σ is the standard deviation of the activity stop time (min).

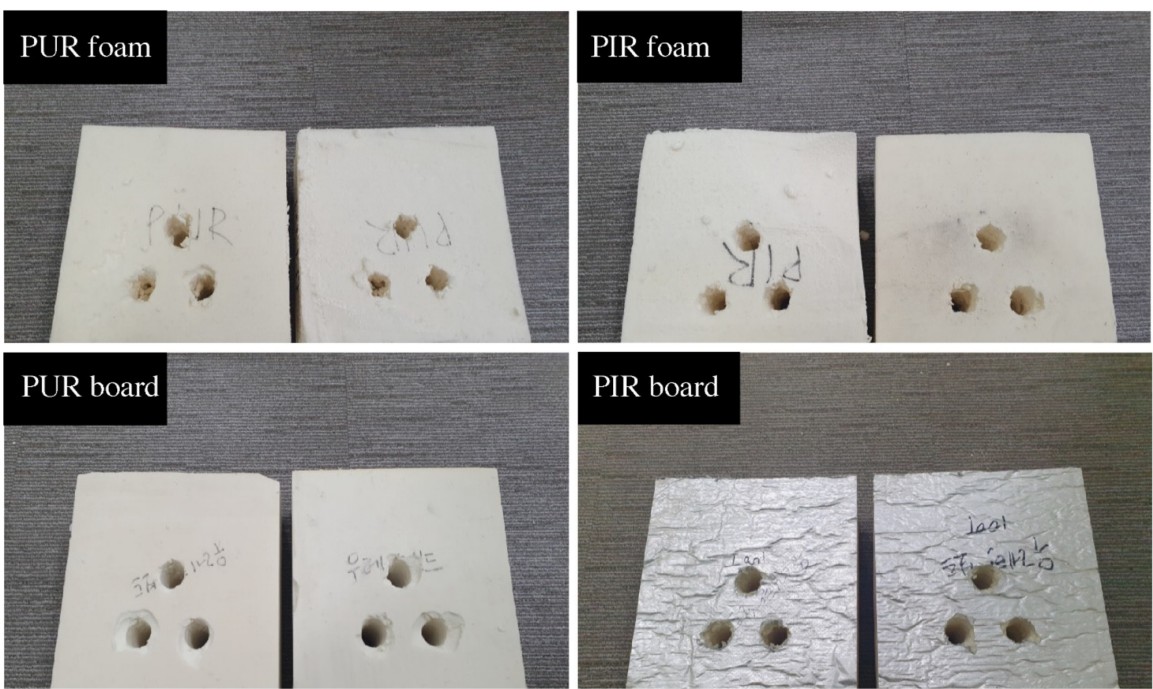

**Figure 5.** Samples used for gas toxicity tests (four types).

If no incapacitation was observed, ⁻X was assumed to equal the total test time (15 min). Otherwise, ⁻X was calculated as

$$^{\overline{}}X = (x\_1 + x\_2 + x\_3 + x\_4 + x\_5 + x\_6 + x\_7 + x\_8)/8 \tag{2}$$

where x (min) is the activity stop time of an individual mouse [13,14].

*2.5. Evaluation of Real-Fire-Scale Flame Propagation Characteristics*

The employed scenario relied on the fact that most polyurethane fires are caused by the ignition of combustible materials on the floor due to their exposure to welding fire during construction, then typically spreading to the ceiling along the wall surface.

The full-scale real-fire test was conducted to compare the fire propagation speeds of different insulating materials in case a fire breaks out from the urethane spray foam or board-type product used to construct the ceiling of a building, with the test room dimensions equaling 3.6 × 1.2 × 2.4 m (Figure 6). The dimensions of the tested sample were halved to reduce the risk of rapid fire propagation in the laboratory. The wall and ceiling of the test room were built from 150-mm-thick urethane foam (PUR, PIR)/urethane board (PUR, PIR). The above test was conducted to quantitatively investigate whether the urethane of the ceiling could be ignited, its ignition time, flame propagation speed, and flashover occurrence time after the ignition of the combustible material at the bottom. As a fire source, a timber crib (40 × 40 × 500 mm, 35 sticks) was installed at the bottom in front of the wall after the partition with the combustible material at the bottom, and ignition was performed using n-heptane.

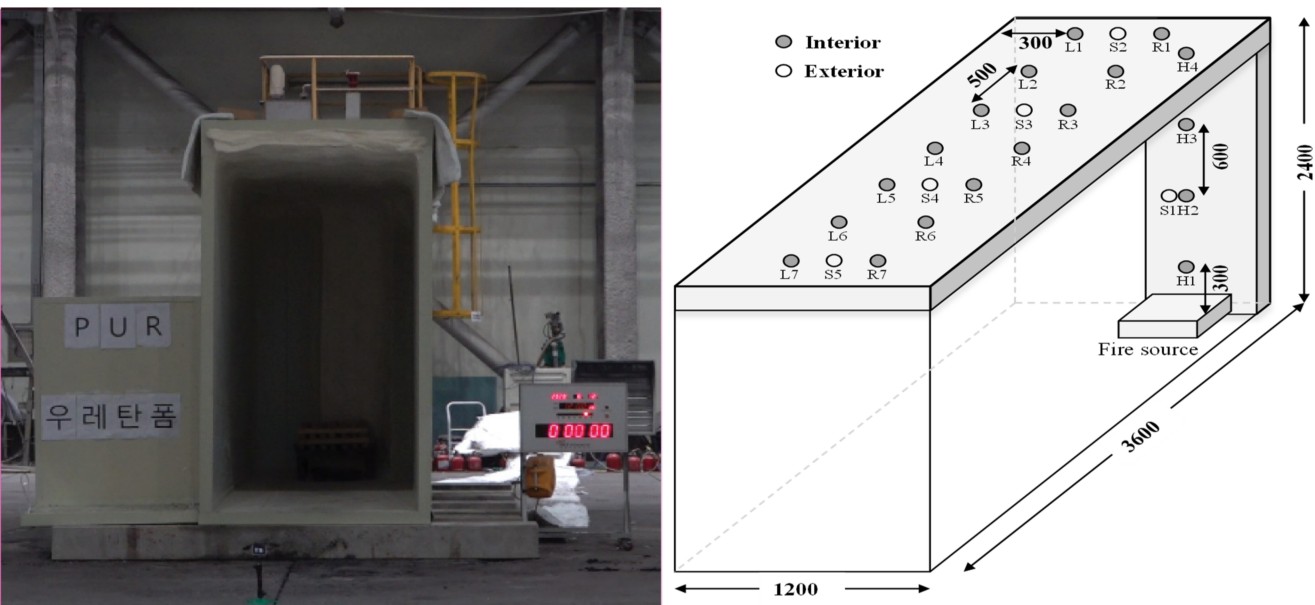

**Figure 6.** Setup of the combustibility test and the employed test room.

The internal and external temperatures were measured inside the insulating material in the upper part of the test room (L1–L7 and R1–R7), outside the upper insulating material (S2–S5), inside the wall insulating material (H1–H4), and outside the wall insulating material (S1). The flame propagation speed was determined from the temperatures recorded by the installed K-type thermocouples.

### 3. Results

*3.1. Cone Calorimeter Tests*

Table 6 shows the results of the cone calorimeter tests, while Figure 7 shows the evolution of HRR with time for different samples. HRR, i.e., the amount of energy released per unit surface area, can be used to assess the risk of fire [15,16] as it represents the maximum fire intensity and can be employed as a proxy for the rate and extent of fire propagation [17,18]. Specifically, high HRR is positively correlated with the danger at the initial fire stage.

**Table 6.** Results of the cone calorimeter tests.

| | PUR F | PIR F | PIR F | PIR B |
|---|---|---|---|---|
| Peak HRR time (s) | 35 | 17.5 | 10 | 165 |
| Peak HRR (kW m$^{-2}$) | 268.75 | 189.89 | 113.89 | 105.80 |
| | PUR F $\gg$ PIR F $\gg$ PUR B $\gg$ PIR B | | | |
| Mean HRR (kW m$^{-2}$) | 66.95 | 73.82 | 46.85 | 22.32 |
| | PIR F $\gg$ PUR F $\gg$ PUR B $\gg$ PIR B | | | |
| THR (MJ m$^{-2}$) | 20.01 | 22.03 | 14.02 | 10.31 |
| | PIR F $\gg$ PUR F $\gg$ PUR B $\gg$ PIR B | | | |

Total heat release (THR) is the overall energy released on the surface of the sample during combustion, reflecting the possibility of flame propagation from the surfaces of other materials [19]. Accordingly, this value is correlated more with the mean HRR than with the peak HRR. It can be said that materials that release energy relatively rapidly pose higher fire risks than those releasing energy slowly [20].

Although PUR F featured a higher peak HRR than the flame-retardant-containing PIR F, the latter material had a higher mean HRR and THR. The PIR F high THR was ascribed

to the slow rate of flame propagation due to the presence of a flame retardant and the resulting persistence of the after-flame, whereas PUR F burned rapidly in the beginning.

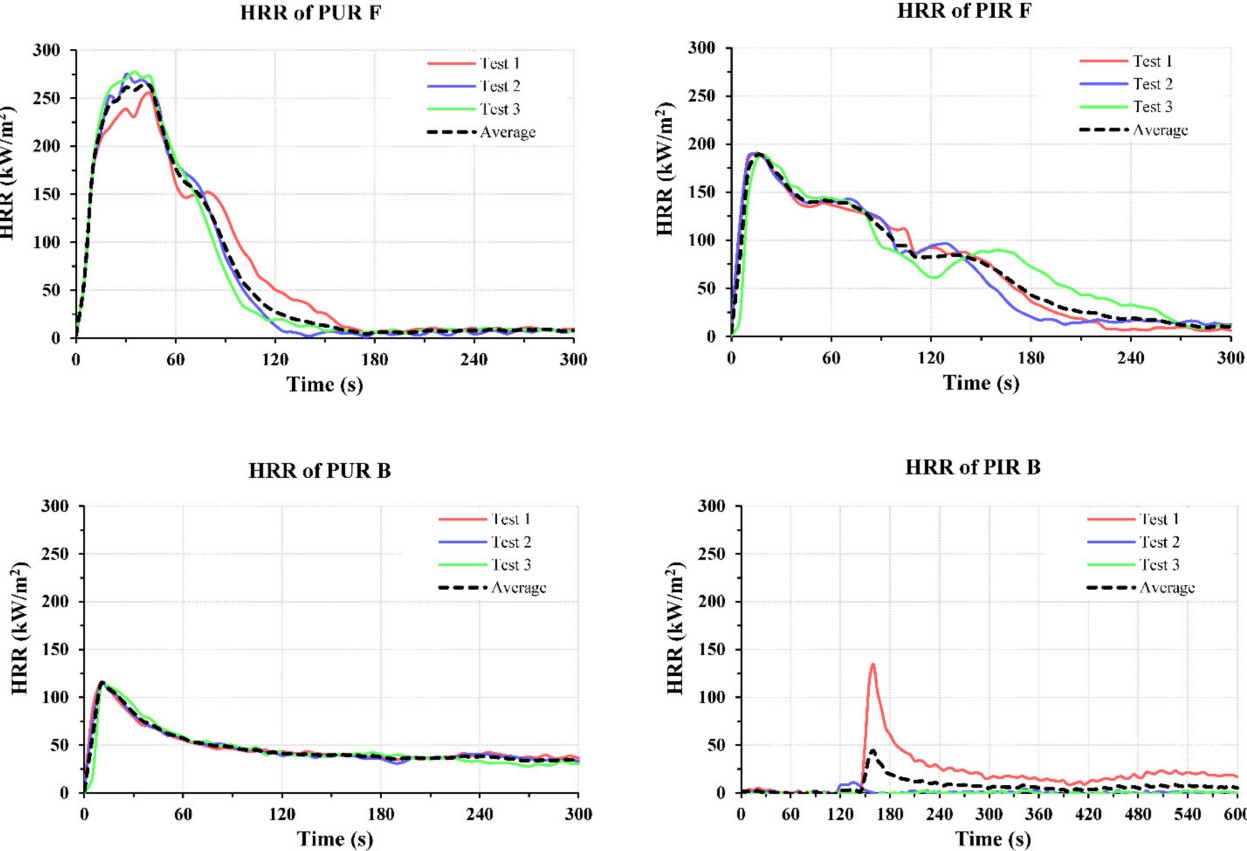

**Figure 7.** Evolution of HRR with times for different polyurethane products.

All the PUR B parameters exceeded those of PIR B, which was ascribed to differences in the surface conditions of these materials. On combustion, the material surface becomes covered with a layer of decomposition products, which have low volatility and are difficult to decompose. Although the state of the layer depends on the polymer type, the solidification degree of the surface layer is generally positively correlated with flame retardancy [21].

As for the limitation of this test, in the case of PIR B, the cases satisfying and not satisfying the criteria occurred one after the other when the test was repeated three times (Figure 8); the cases that failed to satisfy the criteria were compared with those observed for other materials. In the cone calorimeter test, only one end of the specimen was heated, and therefore, the result obtained for the PIR B surface wrapped with aluminum foil was different from that obtained for the surface with no aluminum foil. Accordingly, additional analysis was conducted through full-scale tests.

### 3.2. Gas Toxicity Tests

The gases generated in the combustion chamber flowed to the exposure chamber via the agitation chamber. The combustion test conducted in the combustion chamber revealed that no specimens complied with the safety requirement as they failed to satisfy the flame retardancy criteria. Table 7 lists the results of gas toxicity tests, showing that the second PIR B test featured an aspect different from that of the first test, whereby direct heat was applied to the internal surface because of the contraction of the aluminum foil on the surface (Figure 9). Comparison of PUR F and PIR F revealed that more toxic gas was generated by the latter (flame-retardant-containing) specimen.

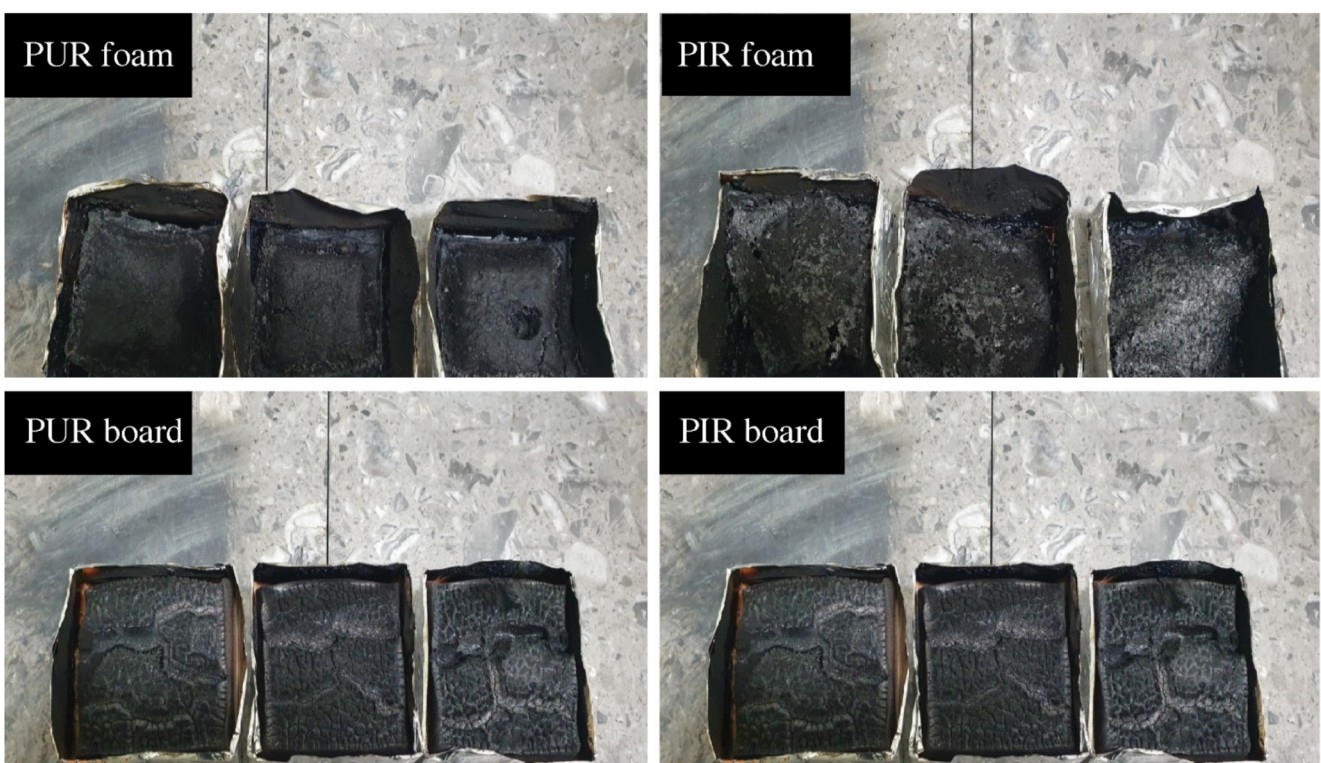

**Figure 8.** Appearance of samples after cone calorimeter tests.

**Table 7.** Results of the gas toxicity tests.

| Sample | 1 | | 2 | |
|---|---|---|---|---|
| | Average Time (m:s) | Standard Deviation (m:s) | Average Time (m:s) | Standard Deviation (m:s) |
| PUR F | 7:16 | 1:50 | 8:10 | 0:38 |
| PIR F | 4:59 | 0:28 | 5:05 | 0:25 |
| PUR B | 6:00 | 0:26 | 4:24 | 0:24 |
| PIR B | 11:03 | 1:30 | 7:27 | 1:02 |

The flame retardants generally used for polyurethane foams are classified into additive-type and reactive-type depending on their usage mode [22,23]. Moreover, flame retardants can also be divided into organic (phosphorus-based, halogen-based, and nitrogen-based) and inorganic (aluminum hydroxide-based, antimony oxide-based, and magnesium hydroxide-based). The flame retardant effect is the result of an interaction between the gas phase and the condensed phase [24]. During the combustion of a material that contains a flame retardant such as bromine or chlorine, toxic gases (HCl, $Br_2$) are emitted together with corrosive, irritating, and toxic smoke [25–28].

Flame retardants considerably affect the emission of non-combustible gases such as $CO_2$, $H_2O$, $NH_3$, and $SO_2$ produced by pyrolysis, which can dilute combustible gases [29] to inhibit spontaneous ignition and continuous combustion [30]. On the other hand, the fire toxicity of PUR was measured to exceed that of PIR under flaming fire conditions because of the presence of a flame retardant in the former case [31].

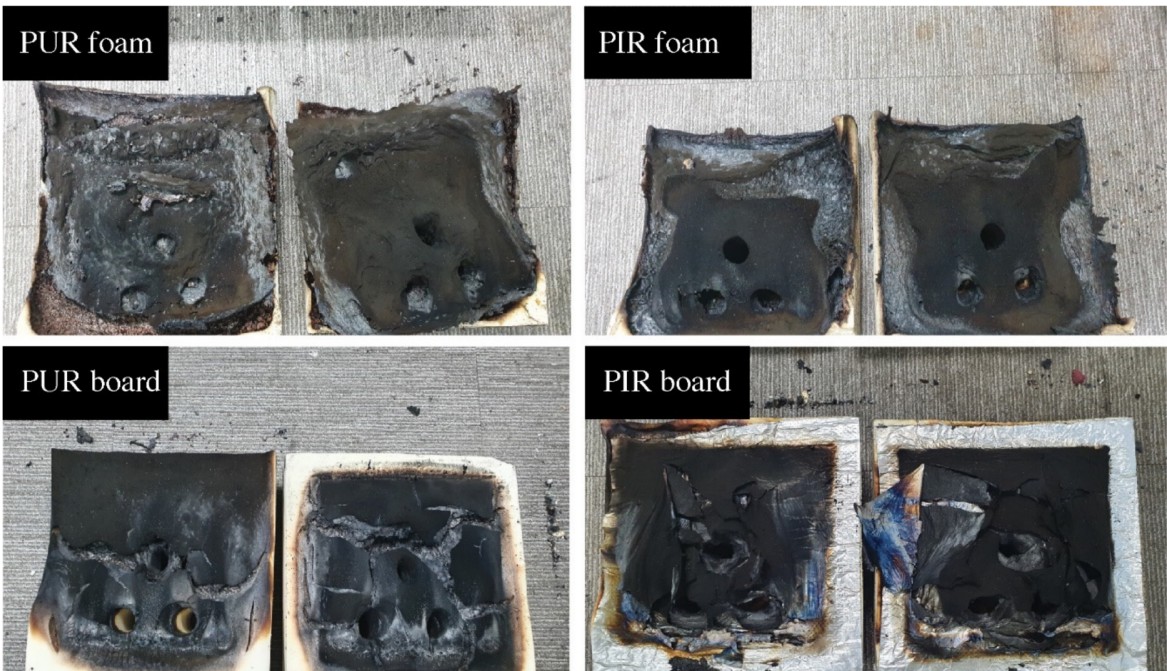

**Figure 9.** Appearance of samples after gas toxicity tests.

The gas toxicity test facilitates evaluation by providing easy-to-interpret results, i.e., the average incapacitation times of test animals. Although the cause of incapacitation for any combustion product can be confirmed, the above test cannot be specifically applied to the flame retardant present in each product. In addition, the causes of incapacitation include a state of being unable to act caused by an anesthetic gas and death caused by toxicity, and quantitative discrimination is not possible. However, unlike the existing toxicity evaluation method that assesses toxicity using the research results for animals, the gas toxicity test allows one to determine the actual effects of the combustion gases on animals and comprehensively identify the damage caused by toxicity due to a single factor, as well as by toxicity due to the combined action of all combustion products [14].

### 3.3. Full-Scale Tests

The full-scale test was performed to confirm the effect of flame retardant addition to the urethane on the time required for the flame to be delivered from the polyurethane surface to the inside. In addition, we also used temperature measurements to compare (i) the rates of flame propagation through the wall to the ceiling after the ignition of the combustible at the bottom, and (ii) the flashover occurrence times.

### 3.3.1. Comparison of Internal and External Temperatures of PUR F and PIR F

The rapid spread of fire on the exterior of both PUR F and PIR F samples made the comparison of the inside and outside walls and ceilings meaningless. However, the flame retardant added to PIR F reduced the amount of heat delivered to the inside. The combustion of foam features the stages of (i) tar formation via foam decomposition, (ii) tar burning in the form of a full burner, and (iii) foam reduction with the consumption of tar and flame development [4]. Visual observations showed that after the ceiling surface was ignited, the foams decomposed to form tar (Figure 10), which flew down in the form of a liquid as soon as it was formed. Some heat emitted from the tar remaining on the surface was re-supplied to foams to accelerate combustion, and the tar flew in the form of charcoal residues at a high temperature [32,33].

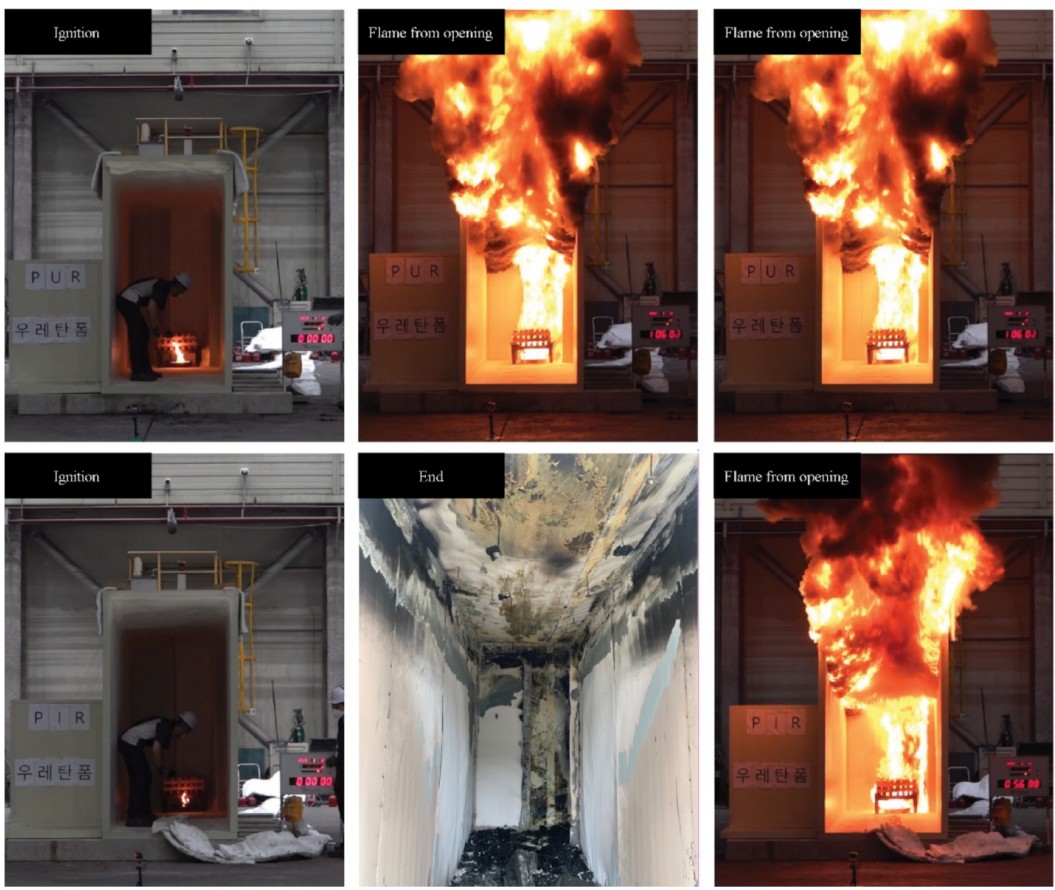

**Figure 10.** PUR F (**top row**) and PIR F (**bottom row**) test rooms at different test stages.

Table 8 and Figure 11 show the temperatures measured inside and outside PUR F and PIR F. In the case of PUR F, the flame ignited on the vertical wall 20 s after the start of the test, the end of the flame reached the ceiling in 60 s, and the flame began to erupt through the opening in an instant 5 s after that. Temperature monitoring afforded the following readings: S1 (exposed on the outside of the ceiling)—625 °C in 64 s, S4—606 °C in 93 s. Moreover, the high speed of flame propagation led the entire sample to reach the flashover stage. Inside the foam, where temperature changes were minor, the following readings were obtained: L1—609 °C in 259 s, L3—602 °C in 192 s, and L5—601 °C in 182 s, which showed that the flame propagated from the opening of the test sample toward the inside.

**Table 8.** Results of internal and external temperature measurements in test rooms made from PUR F (left column) and PIR F (right column).

| State | Point of Information | | PUR F | | PIR F | |
|---|---|---|---|---|---|---|
| | | | Time (s) | Temperature (°C) | Time (s) | Temperature (°C) |
| Flashover (°C) | External | S1 | 64 | 625.2 | 56 | 630.0 |
| | | S2 | 88 | 603.0 | 61 | 610.9 |
| | | S3 | 73 | 618.3 | 76 | 603.4 |
| | | S4 | 93 | 606.3 | 66 | 620.5 |
| | | S5 | -OVER | -OVER | -OVER | -OVER |

**Table 8.** *Cont.*

| State | Point of Information | | | PUR F | | PIR F | |
|---|---|---|---|---|---|---|---|
| | | | | Time (s) | Temperature (°C) | Time (s) | Temperature (°C) |
| Flashover (°C) | Internal | | H2 | 793 | 602.5 | 1341 | 600.0 |
| | | | L1 | 259 | 609.2 | 288 | 603.0 |
| | | | R1 | 346 | 613.4 | 373 | 613.4 |
| | | | L3 | 192 | 602.6 | 422 | 601.8 |
| | | | R3 | 206 | 602.2 | 417 | 605.2 |
| | | | L5 | 182 | 601.8 | 474 | 601.8 |
| | | | R5 | 225 | 607.4 | 493 | 600.6 |
| | | | L7 | 216 | 608.5 | 563 | 600.1 |
| | | | R7 | 307 | 607.1 | 297 | 600.3 |
| Max Temperature (°C) | External | | S1 | 830 | 1135.3 | 1448 | 992.8 |
| | | | S2 | 384 | 1093.4 | 260 | 1108.3 |
| | | | S3 | 375 | 1164.2 | 553 | 1072.9 |
| | | | S4 | 272 | 1021.1 | 238 | 1049.3 |
| | | | S5 | -OVER | -OVER | -OVER | -OVER |
| | Internal | | H2 | 857 | 1093.2 | 1466 | 920.8 |
| | | | L1 | 316 | 970.0 | 355 | 909.5 |
| | | | R1 | 382 | 1049.0 | 723 | 939.8 |
| | | | L3 | 286 | 1020.1 | 737 | 919.2 |
| | | | R3 | 375 | 1158.0 | 495 | 1052.5 |
| | | | L5 | 286 | 976.5 | 756 | 813.8 |
| | | | R5 | 270 | 991.4 | 749 | 754.6 |
| | | | L7 | 294 | 796.4 | 685 | 830.3 |
| | | | R7 | 376 | 782.4 | 426 | 1003.5 |

The results of the cone calorimeter tests showed that the ignition of PIR F was somewhat delayed compared to that of PUR F because of the presence of a flame retardant in the former case. On the other hand, the full-scale test revealed no big difference in the rate of fire spreading. Visual observations showed that the flame was ignited on the vertical wall in 30 s after the ignition, and one end of the flame reached the ceiling part in 50 s, with the flame erupting to the outside 6 s later. These results were no different from those obtained for PUR F. Temperature monitoring afforded the following readings: S1—630 °C in 56 s; S4—620.5 °C in 66 s, and then the flashover was reached.

For the inside, the following temperatures were recorded: 603 °C at L1 in 288 s; 601 °C at L3 in 422 s, and 601 °C at L5 in 474 s, which showed that the flame propagated sequentially from the inside of the test sample toward the opening. The rate at which the flame was delivered to the inside of the foam was reduced by the flame retardant.

3.3.2. Comparison of Internal and External Temperatures of PUR B and PIR B

Visual observations showed that in the case of PUR B, the flame ignited on the vertical wall 20 s after the start of the test, reached the ceiling in 37 s, and began to erupt through the opening 6 s afterward (Figure 12).

Table 9 and Figure 13 show the temperatures measured inside and outside PUR B and PIR B. Temperature monitoring afforded the following results: S2 (exposed outside the ceiling)—634 °C at 55 s; S5—601 °C in 95 s (i.e., when the entire test sample reached the flashover stage). Unlike in the case of fast external propagation, flame propagation inside the test box was slow: R1—606 °C in 445 s; R3—603 °C in 1289 s; R5—600 °C in 1736 s; R7—600 °C in 2489 s. PIR B is produced by attaching a piece of kraft paper on one side and integrating mineral fleece, composite film, or aluminum foil on the other side (aluminum foil is widely used) [34]. The attached aluminum foil offers the benefits of flame retardancy and heat resistance while reducing heat loss. The flame was ignited on the vertical wall 20 s after the start of the test, reached the ceiling in 125 s, and started to erupt through the opening in 178 s, while the aluminum foil began to slowly come off in 150 s and came off completely when the flame began to erupt through the opening. The temperature reached 604 °C in 168 s at S2, 614 °C in 178 s at S3, 612 °C in 192 s at S4, and 605 °C in 202 s at S5, which showed no big difference in the speed of fire spreading compared to the case of PUR

B. In the case of the inside, the temperature was measured as 600 °C at L1 in 1872 s, 601 °C at R2 in 2173 s, 209 °C at L5 in 4315 s, and 70 °C at R7 in 4315 s, failing to reach 600 °C.

### 3.3.3. Comparison of Flame-Propagation Characteristics of Polyurethane Foam and Board

Regarding the fire patterns of PUR F and PUR B, the external flame propagated rapidly irrespective of the material form, whereas large differences in the time at which a temperature of 600 °C was reached were observed in the case of internal flame propagation. These differences were ascribed to the variations in the air inflow due to increased material density, which resulted in a three-fold increase in flame duration. The maximum temperatures inside both samples were measured to have risen above 1000 °C, and all materials excluding the tested sample were burnt out after the test, with only charcoal residues remaining at the bottom.

**Table 9.** Results of internal and external temperature measurements in test rooms made from PUR B (left column) and PIR B (right column).

| State | Point of Information | | PUR B | | PIR B | |
|---|---|---|---|---|---|---|
| | | | Time (s) | Temperature (°C) | Time (s) | Temperature (°C) |
| Flashover (°C) | External | S1 | 57 | 617.2 | 172 | 619.4 |
| | | S2 | 55 | 634.1 | 168 | 604.9 |
| | | S3 | 57 | 610.3 | 178 | 614.0 |
| | | S4 | 528 | 600.0 | 192 | 612.5 |
| | | S5 | 95 | 601.2 | 202 | 605.2 |
| | Internal | H2 | 1191 | 600.1 | 2854 | 601.8 |
| | | L1 | 1043 | 600.0 | 1872 | 600.0 |
| | | R1 | 445 | 606.5 | 2228 | 600.0 |
| | | L3 | 1439 | 602.0 | 2414 | 600.2 |
| | | R3 | 1289 | 603.5 | 2173 | 601.2 |
| | | L5 | 1861 | 601.1 | Not reached | Not reached |
| | | R5 | 1736 | 600.8 | Not reached | Not reached |
| | | L7 | 2418 | 600.4 | Not reached | Not reached |
| | | R7 | 2489 | 600.3 | Not reached | Not reached |
| Max Temperature (°C) | External | S1 | 553 | 932.0 | 3170 | 1023.8 |
| | | S2 | 554 | 1003.8 | 513 | 879.9 |
| | | S3 | 1349 | 946.4 | 2007 | 955.1 |
| | | S4 | 2174 | 751.9 | 212 | 752.4 |
| | | S5 | 2672 | 1011.1 | 213 | 632.3 |
| | Internal | H2 | 1814 | 1134.5 | 3190 | 1152.7 |
| | | L1 | 1116 | 1181.2 | 2423 | 803.7 |
| | | R1 | 688 | 1038.0 | 3450 | 861.5 |
| | | L3 | 1562 | 1021.9 | 2543 | 660.3 |
| | | R3 | 1431 | 1102.6 | 2385 | 936.5 |
| | | L5 | 2287 | 901.2 | 3525 | 208.0 |
| | | R5 | 2027 | 854.8 | 4792 | 215.1 |
| | | L7 | 2967 | 893.1 | 4786 | 39.5 |
| | | R7 | 2715 | 1081.0 | 4803 | 75.2 |

In the cases of PIR F and PIR B, the external flame propagated rapidly in the beginning. In the case of PIR F, the flame propagated to the inside and thus raised the temperatures of all the thermocouples above 1000 °C. In the case of PUR F, the flame stopped without propagating to the opening part, excluding the part directly above the combustible at the bottom. As explained in Section 2.1, the isocyanurate ring decomposes at 450–650 °C, while in the case of PIR B, the internal temperature failed to rise above 200 °C, and combustion, therefore, stopped. As for PIR B, traces of local combustion were found only in the place with a close separation distance inside the ceiling, and the opening part maintained its shape. Flame retardants delay pyrolysis, reduce the temperature rise on the material surface, and diffuse the heat flowing in the flame over the entire material [35]. Moreover,

flame retardants hinder energy transfer between the solid in the combustion area and the combustible gas by forming char before combustion and feature self-extinguishing properties [36,37].

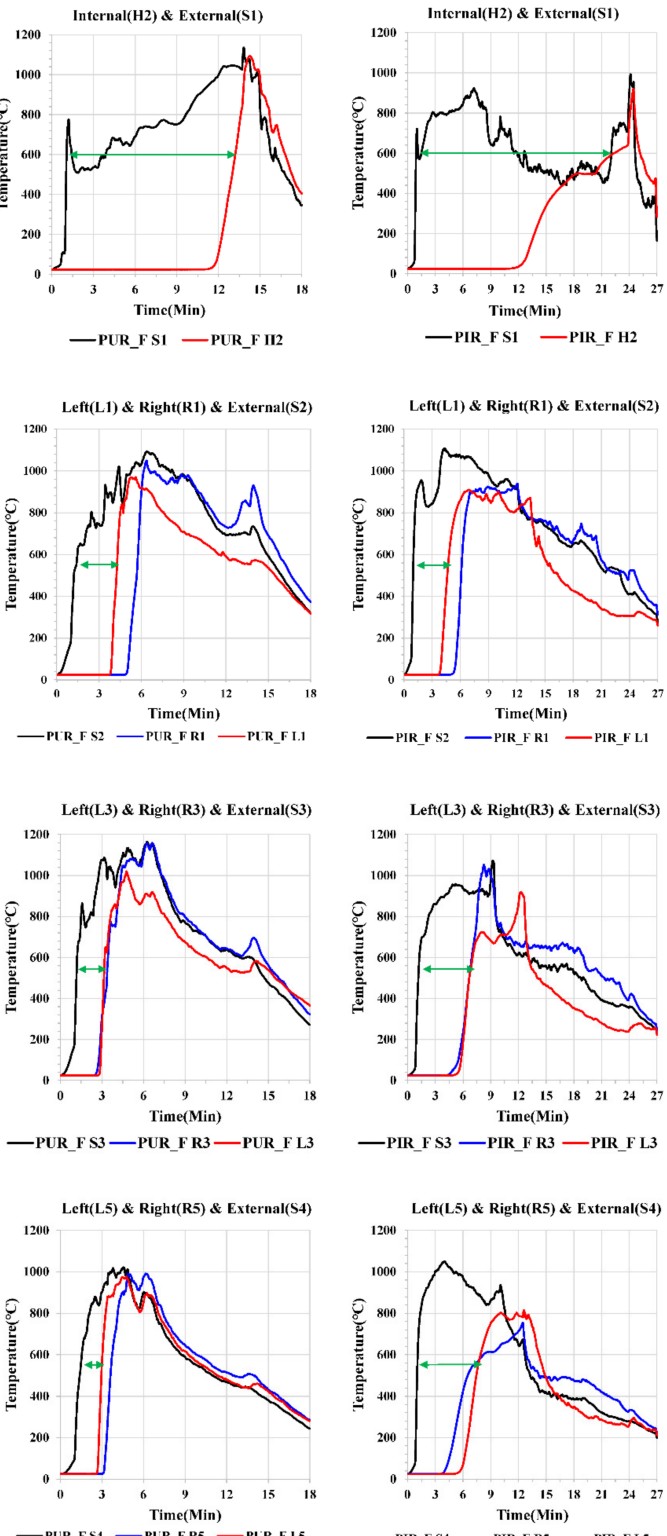

**Figure 11.** Results of internal and external temperature measurements in test rooms made from PUR F (**left column**) and PIR F (**right column**).

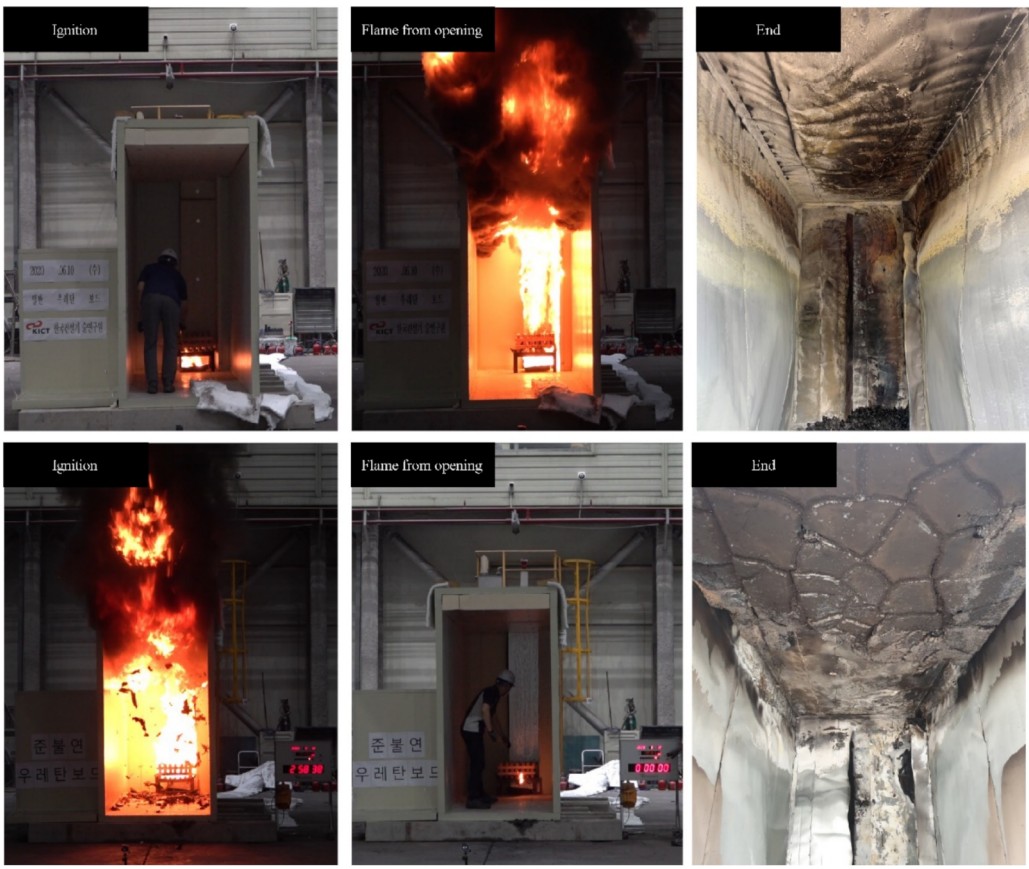

**Figure 12.** PUR B (**top row**) and PIR B (**bottom row**) test rooms at different test stages.

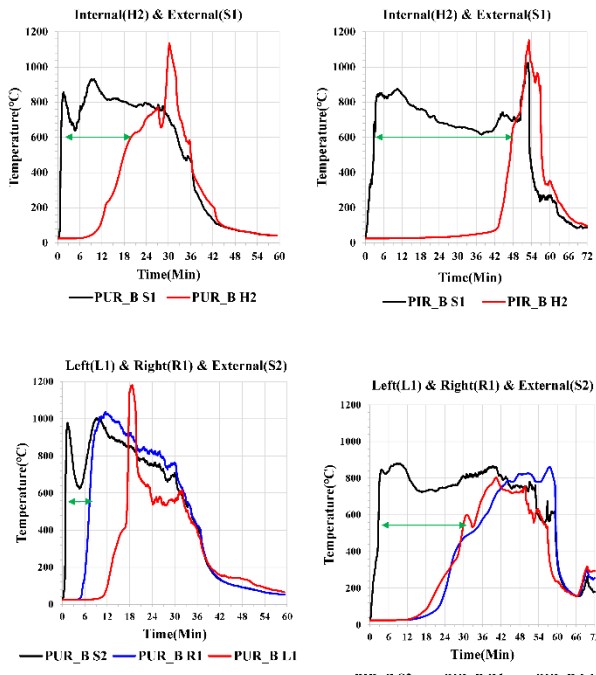

**Figure 13.** *Cont.*

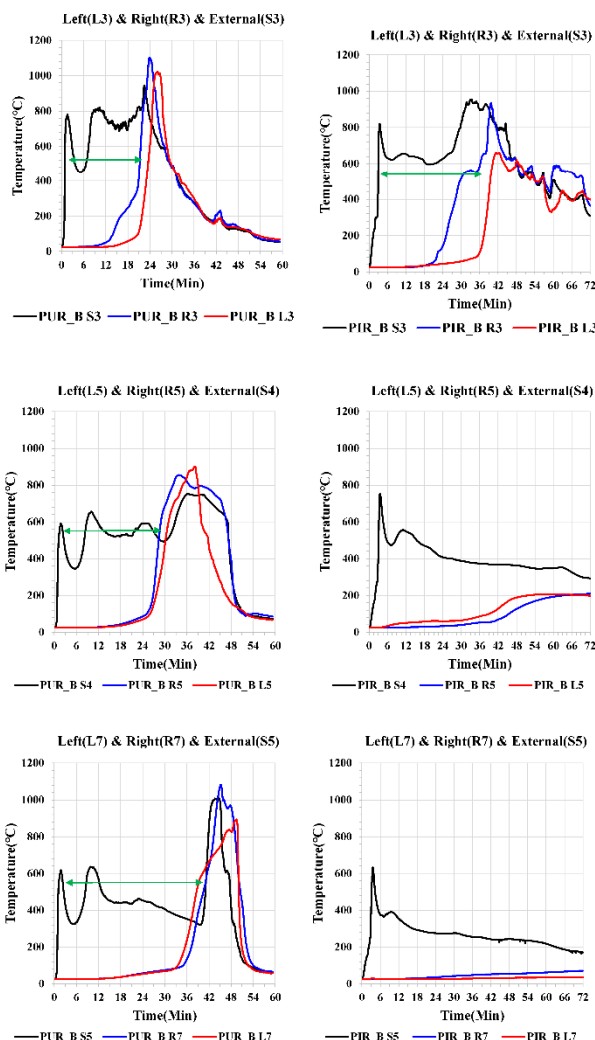

**Figure 13.** Results of internal and external temperature measurements in test rooms made from PUR B (**left column**) and PIR B (**right column**).

## 4. Discussion

The flame-propagation properties and fire risks of four foam- and board-type polyurethane insulation materials used in internal and external finishings were examined, and the following conclusions were drawn:

(1) Except for PIR B (which conformed to flame retardancy criteria but not to semi-non-combustibility criteria), the samples did not satisfy the flame-retardancy criteria of KS F ISO 5660-1. The total heat released was in the order of PUR F > PIR F > PUR B > PIR B. While PUR F and B rapidly combusted in the beginning, PIR F and B (impregnated with a flame retardant) featured slow flame propagation and long after-flame persistence times;

(2) The average incapacitation time, which is an index of toxicity, was shorter for PIR F than for PUR F and PUR B because of the toxic gases (HCl and $Br_2$) emitted by the flame retardant. On the other hand, fire toxicity was high under flaming fire conditions;

(3) The propagation of the external flame was accompanied by a flashover irrespective of whether a flame retardant was added or not. The maximum flame propagation temperatures inside PUR F and B and PIR F were 1000 °C or higher. Visual observation conducted after the occurrence and end of flashover showed that the entire specimen burned, with only a pile of charcoal residues remaining at the bottom. In the case

of PIR B, combustion stopped without propagating to the vicinity of the opening (excluding the part directly above the combustible at the bottom);

(4) The flame retardant delayed/blocked the transfer of energy to the inside. When a flame reaches the ceiling, it rapidly spreads, at which time the presence of a combustible gas greatly affects the flame propagation speed. Covering the specimen with a non-combustible inorganic material or a flameproof coating was shown to effectively delay flame propagation in the beginning;

(5) The attachment of aluminum foil and the application of an inorganic material on the surface effectively prevented ignition. The rapid propagation of the flame on the surface in the beginning was found to have the ability to cause a second ignition of a combustible material far away from the flame, which suggested that a full-scale test was required for validation.

## 5. Conclusions

The fire risk posed by a given material is largely determined by its combustion heat. According to previous studies, rigid urethane foams do not burn faster than other materials in terms of mass loss or heat release. However, when a fire breaks out, it spreads rapidly in the case of spray-type materials due to their non-uniform surface, whereas in the case of board-type materials with uniform surfaces, fire can still rapidly propagate due to the powder or micro-holes generated during construction.

Although polyurethanes pose high fire risks because they are highly combustible and feature high flame propagation speeds, they are widely used. Accordingly, when employed in construction, polyurethanes are impregnated with organic and/or inorganic flame retardants to reduce the risk of fire. Herein, temperature measurements showed that flashover occurred irrespective of whether a flame retardant was added or not. However, we note that (i) wood with a relatively low heat release rate was used for the test, and (ii) we failed to present the heat release rate of polyurethane resulting from an oil fire and other flammables at the bottom. Hence, we plan to conduct additional flame propagation tests for different types of flame retardants in the future.

**Author Contributions:** Conceptualization, D.M.C.; methodology, D.M.C. and S.U.C.; formal analysis, E.H.H.; investigation, J.H.C.; data curation, E.H.H.; writing—original draft preparation, J.H.C.; writing—review and editing, J.H.C. All authors have read and agreed to the published version of the manuscript.

**Funding:** This research received no external funding.

**Institutional Review Board Statement:** Not applicable.

**Informed Consent Statement:** Not applicable.

**Data Availability Statement:** Data sharing is not applicable.

**Conflicts of Interest:** The authors declare no conflict of interest.

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
