# Peer review of "Fire Propagation Characteristics and Fire Risks of Polyurethanes: Effects of Material Type (Foam & Board) and Added Flame Retardant"

_fire, doi:10.3390/fire5040105_

Round 1
Reviewer 1 Report
the article analysis combustion properties of polyurethane and polyisocyanurate foams and boards. The article would be interesting and of great importance if it is correctly formatted and written. I have few remarks which are presented below:
1) Materials and methods part (the text before 2.1 section and the text in 2.1 section) is not properly written. The information presented in this section is considered as a literature review and/or discussion so it should be placed rather in Introduction section or Results and Discussion section.
2) The products used for the test were not described properly.
3) Why toxic gases such as CO2 and CO which are released during combustion were not analysed? I mean, the amount released.
4) SEM images after combustion tests are also of great importance as they show the char layer and its homogeinity.
5) Article does not contain Discussion part. Authors should consider writing Results and Discussion section or do it separately, i.e. Results first and then Doscussion.
6) What about upper and lower limits of the results presented in Tables? Or at least standard deviations?
Author Response
Thank you very much for the reviewer's comments.

Reviewer 2 Report
The subject of the manuscript focused on fire propagation characteristics and fire risks of polyurethanes and the effects of material type (foam vs. board) and added flame retardant is in good relevance with the scope of FIRE.
The introduction correctly presents a review of publications in the area of research presented in the manuscript. The methods used are described with all necessary data. Discussion of the results conducted correctly. The conclusion concerning the flame propagation properties and fire risks of four foam- and board-type polyurethane insulation materials used in internal and external finishings indicate a clear effect of the type of material and the type of flame retardant added on the flammability of these materials . The entire manuscript is correctly written and does not require changes or additions.
However, the section devoted to the description of the materials used requires a significant addition. It describes the types of materials used, which has a significant impact on their flammability. It is necessary to complete the information about the trade name of the products used and the name of the manufacturer. These information will help readers who are interested in a possible verification of the results presented in the manuscript.
Author Response

(The authors gave the same response as above.)

Reviewer 3 Report
1. Please explain what ’average incapacitation times of mice’ refer to on page 3 line 79.
2. Lines 144 and 145 states that in the cone calorimeter test, radiant heat is applied at a heat flux of 50 kW/m2. Does this only apply to your test? If yes, please clarify since the irradiance from the cone can range from 0 – 100 kW/m2.
3. What is the reason for creating three holes in the samples for the gas toxicity test?
4. Please elaborate on the differences between the surfaces covered with aluminum foil and that with no foil after burning.
5. Please provide the time to ignition of the samples tested in the cone calorimeter.
6. I will suggest that the authors put the temperature readings for the full scale tests in a table including the time and temperature flashover occurred for the samples.
Author Response

(The authors gave the same response as above.)

Reviewer 4 Report
Dear authors, the article is certainly relevant.
PIR and PUR are widely used in construction all over the world, and not only in Korea. Such problems are not only relevant for Korea, unfortunately, and happen all over the world, because PPU is very widespread. And fire safety during construction and welding work is necessary and this applies to all countries. There are several comments on the article.
1. The abstract should pay attention to the world practice in order to make it interesting for all the readers of the world, and then go to Korea as an example.
2. The authors submit the article to a scientific journal, so in the title and in the text (e.g., 3.3.2) it is necessary to avoid popular combinations such as "vs". The title can be of the type "Reduction of fire hazard of PUR and PIR for building envelopes and roof coverings" (e.g.) "Comparative analysis of results..."
3. Did the authors come up with Figure 1 themselves? Or was it taken from a literature source?
4. In the "Introduction" section you should add the relevance of the study, add sources of literature, purpose, objectives and comparison of your study with previous studies. There have been a lot of people doing PPU combustion. It is strange that the «Introduction» section contains only TWO literature sources. The main body of sources should be in the "Introduction" section.
5. Line 67-72 - add literature and it should all be in the "Introduction" section.
6. Line 85 - 144. Obvious items, they can be shortened and this should also be in the Introduction section.
7. Line 73-75. Tell us what PUR and PIR were chosen as samples (density, cell type, strength, rigidity, moisture absorption, combustibility class, vapor permeability, etc.). Each PUR and PIR is made according to standards (for example, EN 14315-1:2013. There are many of them, be specific about your samples, otherwise your results have very poor practical value.
8. Figures 7, 11, 13 do not explain the curves depending on the physical characteristics of the samples, there is no explanation of the discrepancy in the results.
Author Response

(The authors gave the same response as above.)

Round 2
Reviewer 1 Report
Authors have taken into consideration all my remarks.
Author Response
Thanks to the reviewer's opinion, it became a completed manuscript.
Reviewer 4 Report
Dear authors,
1. work has been done, however, you have agreed to change the title and paragraphs where opposites are used not scientifically, but popularly (for example, paragraph .3.3. Foams & boards ).
It is the same in the title and in some paragraphs of the article.
2. table 1 looks strange, because it contains a lot of repetitions. It should either be put in text or combined in the same parameters for the selected materials.
3. What kind of fire retardant did you use? What class? How will other authors refer to your study, if it is not clear what kind of flame retardant and how much of it was in the formulation? The study then doesn't make much sense to science.
Author Response
Point 1: work has been done, however, you have agreed to change the title and paragraphs where opposites are used not scientifically, but popularly (for example, paragraph .3.3. Foams & boards ).
Response 1: Thank you very much for the reviewer's comments. We agree with the reviewer's opinion. The title and paragraph have been corrected throughout the article.
Point 2: Table 1 looks strange because it contains a lot of repetitions. It should either be put in text or combined with the same parameters for the selected materials
Response 2: We agree with the reviewer's opinion. Tables were rewritten by combining the same parameters.
Point 3: What kind of fire retardant did you use? What class? How will other authors refer to your study, if it is not clear what kind of flame retardant and how much of it was in the formulation? The study then doesn't make much sense to science.
Response 3: Thank you so much for reviewing our manuscript. The Polyurethane mixing ratio was not presented because it was difficult to disclose due to the company's security issue. However, the contents of identifiable substances were added. I hope you understand.
Finally, the reviewer's opinion was actively reflected and revised.
Once again, thank you so much for reviewing.
